# Estimate of energy loss from internal solitary waves breaking on slopes

Kateryna Terletska[1], Vladimir Maderich[1], and Tatiana Talipova[2]

[1]Institute of Mathematical Machine and System Problems, Glushkov av., 42, Kyiv 03187, Ukraine
[2]Institute of Applied Physics RAS, Nizhny Novgorod,Russian Federation

**Correspondence:** Terletska K. (kterletska@gmail.com)

**Abstract.** Internal solitary waves (ISWs) emerge in the ocean and seas in various forms and break on the shelf zones in a variety of ways resulting in intensive mixing that affects such processes as biological productivity and sediment transport. As ISW of depression propagates in a two-layer ocean from the deep part onto a shelf two mechanisms are significant. The first is the breaking of internal waves over bottom topography when fluid velocities exceed wave phase speed that causes overturning

of the rear face of the wave. The second is changing of polarity at the turning point where the depths of upper and lower layers are equal. We assume that parameters that described the process of interaction of ISW in a two-layer fluid with the idealised shelf-slope are (i) the non-dimensional wave amplitude, normalized on the upper layer thickness, (ii) the ratio of the height of the bottom layer on the shelf to the incident wave amplitude, and (iii) angle of bottom inclination. Based on a proposed three-dimensional classification diagram four types of wave propagation over the slopes are distinguished: ISW propagates over

slope without changing polarity and wave breaking; ISW changes polarity over slope without breaking; ISW breaks over slope without changing polarity; ISW both breaks and changes polarity over the slope. The energy loss during ISW transformation over slopes with a various angles was estimated using the results of 85 numerical experiments. "Hot spots" of high levels of energy loss were highlighted for idealized bottom configuration that mimics continental shelf at Lufeng Region in the South China Sea.

**Keywords.** Internal solitary waves of depression, wave shoaling, energy loss

## 1 Introduction

Observations demonstrate the evidence of internal solitary waves (ISWs) in coastal oceans and seas (Apel et al., 1995). It is generally accepted that one of the main causes for the occurrence of internal solitary waves is barotropic tide interacting with the bottom topography (Maxworthy, 1979; Gerkema and Zimmerman, 1995).

Generated by tides ISWs of depression (the upper layer thickness is usually much less than the depth of the ocean) are the most energetic and can propagate thousands of kilometers from the origin (Kunze et al., 2012). As the result, ISWs transport the energy far from the location of their generation. Like surface waves, internal waves break at the coastal zone of the ocean. Such waves are an important component of mixing and energy dissipation in the ocean (Liu et al., 1998; Davis et al., 2020). The breaking of ISWs upon sloping boundaries in the coastal region also plays an important role in the diapycnal mixing

(StLaurent et al., 2012), the biological enhancement (Sangra et al., 2001; Wang et al., 2007), resuspension of bottom deposits (Pomar et al., 2012; Boegman and Stastna, 2019).

The ISW's interaction behavior depends on the steepness of the topography and the characteristics of the solitary waves (Garrett and Kunze, 2007). If the slopes are smooth much of the energy scatters upslope onto a continental shelf where it will dissipate but for steeper slopes energy will reflect and return into the deep ocean (Klymak et al., 2010). It is important to under-
stand the mechanisms of transformation of the ISWs at continental slopes and identify "hot spots" of wave energy dissipation. Two shoaling mechanisms can be important: (i) the conversion of the ISWs of depression into elevation waves in a two-layer stratification. When the thickness of the upper mixed layer is greater than one-half the total water depth (Helfrich and Melville, 1986; Cheng et al., 2011; Bai et al., 2021). (ii) ISW breaking on the slope occurs when fluid velocities in the wave exceed wave phase speed. Leading to the overturning of the rear face of the wave, shear instability, and intensive mixing. Different types
of breaking are commonly distinguished by slope inclination, water column stratification, and wave characteristics. Assuming analogy with surface waves the breaking regimes of ISW in a two-layer fluid over slope were classified (Aghsaee et al., 2010; Boegman et al., 2005) into surging, plunging, collapsing, and fission. In these studies, classification of breaking is based on the Iribarren number a ratio of the slope to the square root of the wave steepness (amplitude divided on the wavelength). This criterion was modified by (Nakayama et al., 2019) for collapsing and plunging breakers using of a new wave Reynolds number
that takes into account nonlinear wave steepening. A simple three-dimensional $\alpha\beta\gamma$ classification diagram was proposed by Terletska et al. (2020) to distinguish different regimes of interactions of ISW with the slope-shelf topography. The classification is based on three parameters: the slope angle $\gamma$, the non-dimensional wave amplitude $\alpha$ (wave amplitude normalized on the upper layer thickness), and the blocking parameter $\beta$ that is the ratio of the height of the bottom layer on the shelf to the incident wave amplitude.
ISWs breaking over slopes were observed in many coastal locations over the globe (New and Pingree, 1990; Alford et al., 2015; Vlasenko et al., 2014; Osborn et al., 1980; Nam and Send, 2010; Fu et al., 2016; Orr and Mignerey, 2003; Klymak et al., 2006). Observational studies have shown that amplitudes of depression ISWs in the South China Sea (SCS) could reach extreme values up to two hundred and more meters (Huang et al., 2016; Ramp et al., 2010; Klymak et al., 2010). Based on the analysis of satellite images (Wang et al., 2013; Jackson, 2004) it was found that in the northeastern South China Sea most
internal waves are generated at the Luzon Strait. Further, solitary waves propagate westward and then they diffracted around the Dongsha Islands. In the shallow water regions of the northern SCS, changes in water depth may cause polarity conversion, leading to the transformation of depression ISW to elevation ISWs (Liu et al., 1998). Orr and Mignerey (2003) showed that the kinetic energy of ISWs decreased three times, after changing polarity, Zhang et al. (2018) showed that the seasonal variations in stratification cause these seasonal variations in polarity. The present study focused on the ISWs transformation over an
idealised shelf-slope topography with a two-layer stratification. The objectives are (1) to compare $\alpha$ $\beta$ $\gamma$ classification with the results of numerical modelling, laboratory studies and field observations, (2) to identify high energy dissipation zones of ISWs passed over the shelf-slope topography using the $\alpha$ $\beta$ $\gamma$ classification (3) to apply $\alpha\beta\gamma$ classification to the results of numerical modelling that mimics ISW transformation over a continental shelf at Lufeng Region SCS, and to determine energy loss in result of the transformation of ISWs over the shelf-slope topography. Information about polarity change criteria and

criteria of breaking in $\alpha\beta\gamma$ classification of regimes of ISWs transformation over shelf topography is presented in Section 2. The overview of field and laboratory measurement data and their comparison with the numerical modelling data are given in Section 3. Dissipation of energy of ISWs breaking over shelf topography is considered in Section 4. The results are summarized in the Conclusions.

## 2 Regimes of ISW transformation over slope-shelf topography

A two-layer approximation is a simple model of the stably stratified oceans and lakes. In this model, we approximated stratification by the two continuous layers of depths $h_1$ (upper layer) and $h_2$ (lower layer) with relatively thin pycnocline. When $h_1 > h_2$ internal solitary waves propagate in the form of elevation ISW, and if $h_1 < h_2$ they propagate in the form of the waves of depression. In this study, we consider ISW of depression (with amplitude $a_{in}$ ) propagating over an idealized slope-shelf with a slope $\gamma$ and the minimum depth of the lower layer over the shelf $h_{2+}$. Idealised shelf-slope topography is shown in Fig. 1 b, whereas the idealised configuration that mimics continental shelf at Lufeng Region (Fig. 1 a) SCS is shown in Fig. 1 c .

It was assumed that ISW transformation over a slope is controlled by stratification, slope inclination and amplitudes (wavelength) of the incident wave (Terletska et al. (2020)). Two possibilities that could occur with the wave during shoaling were determined: (i) ISW breaking, which was associated with gravitational instability due to the wave overturning and shear instability, and (ii) changing ISW polarity on the slope.

Three parameters $\alpha$ $\beta$ $\gamma$ can be important for behaviour of the incident wave on slope-shelf (fig. 1 b,c):

1. Slope inclination $\gamma$ (measured as an angle);

2. Blocking parameter $\beta$ that is the ratio of the height of the minimum depth of the lower layer over the shelf $h_{2+}$ (Fig. 1 b,c) to the incident wave amplitude $a_{in}$

$$\beta = h_{2+}/|a_{in}|. \tag{1}$$

3. Nonlinearity parameter that is the ratio of the incident wave amplitude to the depth of upper layer:

$$\alpha = |a_{in}|/h_1. \tag{2}$$

The idea for blocking parameter $\beta$ comes from numerical and laboratory experiments as the "degree of blocking", which is an important parameter that controlling the loss of energy into transmitted and reflected waves passing the obstacle (Vlasenko and Hutter, 2002; Wessels and Hutter, 1996). Parameter $\beta$ was modified by Talipova et al. (2013) where ISWs (as depression and elevation type) passing over step ($\gamma = 90^o$) were considered. It was shown that for the ISW of depression transformation over underwater step is weak for values $\beta > 3$ (when the dynamics of ISW could be described by weakly nonlinear theory), for $2 < \beta < 3$ interaction is moderate (when the main mechanism for ISW breaking over bottom step produce shear instability). For $0.4 < \beta < 2$ the interaction is strong with maximal energy loss, when ISW produced the flow that results in a jets and

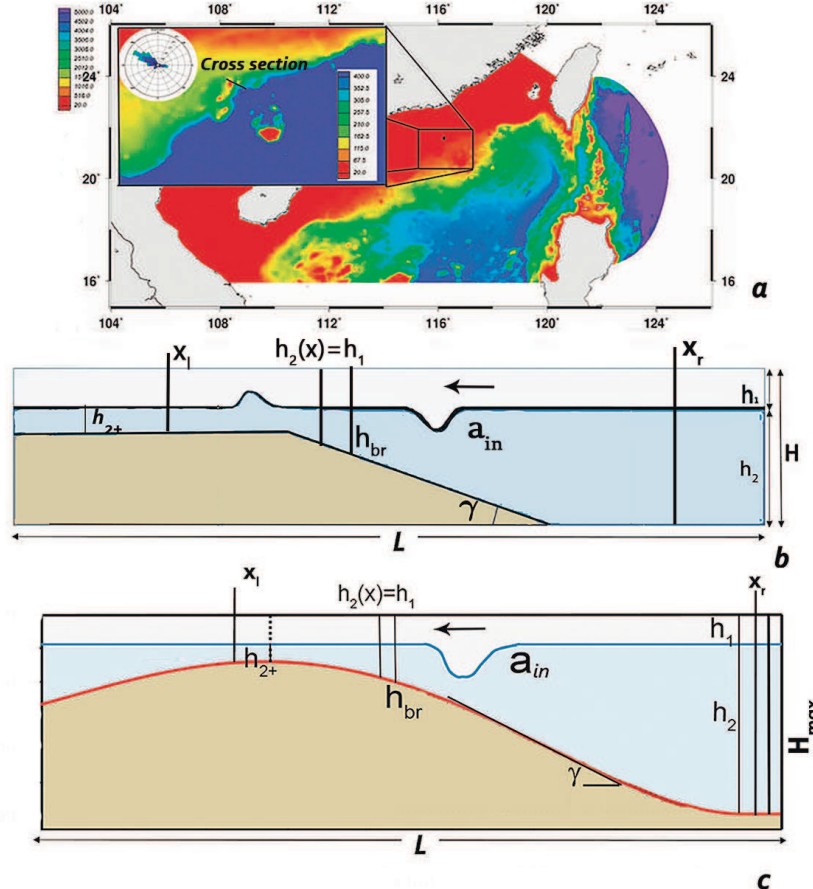

**Figure 1.** Sketch of transformation of depression ISW over a slope-shelf. (a) - SCS with Lufeng Region, (b) sketch of breaking and changing the polarity of ISW of depression after passing through a turning point, (c) - Idealised topography at Lufeng Region (SCS).

vortices. The interaction in the range $-0.9 < \beta < 0.4$ was called 'transitional regime', when the step height between strong interaction and full reflection from the step whereas for $\beta < -0.9$ full reflection from the underwater step takes place.

Internal waves in a framework of weakly nonlinear theory change their polarity in the point where the upper and lower layers are equal (Grimshaw et al., 2004). Notice that numerical experiments using full Naiver-Stokes equations (Maderich et al., 2010) confirm the applicability of the Gardner equation to predict turning point $h_1 = h_2$ even for the wave of large amplitude. This relation for turning point can be expressed through parameters and using

$$\beta = 1/\alpha. \tag{3}$$

**Table 1.** Parameters $\alpha \ \beta \ \gamma$ of ISW in numerical, laboratory experiments and field observations

| Location | zone | $\alpha$ | $\beta$ | $\gamma$ |
|---|---|---|---|---|
| Celtic Sea(Vlasenko et al., 2014) | 3 | $0.8 - 3.3$ | 1.4 | $3°$ |
| Bay of Biscay(New and Pingree, 1990) | 1, 3 | $1 - 2$ | $0.8 - 1.5$ | $4.5°$ |
| Andaman Sea(Osborn et al., 1980) | $0.75 - 1$ | 0.16 | 1 | $0.33° - 1.5°$ |
| Oregon shelf(Moum et al., 2003) | 1 | 0.85 | 4.3 | $0.3°$ |
| South China Sea(Orr and Mignerey, 2003) | 1 | $0.8, 1.55$ | 1.1 | $1°$ |
| South China Sea (Dongsha Atoll)(Fu et al., 2016) | 4 | 1.25 | 0.2 | $3°$ |
| Huntington Beach(Nam and Send, 2010) | $1, 3, 4$ | 0.83 | 1.28 | $0.23°$ |
| East sea(Navrotsky et al., 2004) | 2 | 0.5 | 1 | $0.1°$ |
| St. Lawrence Estuary(Bourgault et al., 2007) | 4 | 1 | 0 | $3°$ |
| Laboratory(Helfrich and Melville, 1986) | 1, 4 | $0.12 - 0.18$ | $0.2 - 5$ | $4°$ |
| Laboratory(Chen, 2007) | 2, 4 | $0.2 - 0.7$ | 0.4 | $14°$ |
| Numerical experiments(Talipova et al., 2013) | $1 - 4$ | $0.2 - 2.2$ | $-2 - 8$ | $90°$ |
| Numerical experiments(Aghsaee et al., 2010) | 4 | $0.2 - 2.$ | 0 | $0.5° - 17°$ |
| Numerical experiments(Terletska et al., 2020) | $1 - 4$ | $0.25 - 1.5$ | $0 - 2.5$ | $0.5° - 90°$ |

For breaking point criteria was taken criterion proposed by Vlasenko and Hutter (2002). It was built based on the Navier–Stokes numerical model simulations data. It was found that the ratio of the amplitude of the incident wave $a_{in}$ into the value of the undisturbed thickness of the lower layer in point where the wave breaking takes place $h_{br}$ (Fig. 1 b,c), is the function of the slope $\gamma$:

$$\frac{|a_{in}|}{h_{br}} = \frac{0.8^o}{\gamma} + 0.4. \tag{4}$$

For each slope angle $\gamma$ the blocking parameter value $\beta_{br}$ that divide zone of the non-breaking regime for $\beta > \beta_{br}$ and breaking regime for $\beta < \beta_{br}$ can be found from (3) and (4) at $h_{2+} = h_{br}$ that

$$\beta_{br} = \gamma/(0.8^o + 0.4\gamma). \tag{5}$$

We can also obtain value $\alpha_{br}$ that divide zone 4 on breaking regime when ISW first breaks $\alpha > \alpha_{br}$ and when wave first changes polarity and then breaks $\alpha < \alpha_{br}$. It can be found from(3) using (5) that yields

$$\alpha_{br} = (0.8^o + 0.4\gamma)/\gamma. \tag{6}$$

Thus four different scenarios of internal solitary wave interaction with a shelf-slope topography in a two-layer approximation can be realized: (1) non-breaking regime without changing polarity, (2) - non-breaking regime with changing polarity, (3) breaking regime without changing polarity, (4) - breaking regime with changing polarity.

110      Analysing equations (3) (5) we conclude that parameters $\alpha, \beta, \gamma$ control the processes for both the wave breaking and the wave polarity change. A three-dimensional diagram with the dependence on parameters $\alpha, \beta, \gamma$ ($\alpha\beta\gamma$ diagram) is given in Fig. 2 showing four zones: (zone 1) ISWs transform without changing polarity and wave breaking; (zone 2) ISWs transform with changing polarity without breaking; (zone 3) ISWs break without changing polarity; (zone 4) ISWs break with changing polarity. In the space of $\alpha, \beta, \gamma$ these regimes are separated by the surfaces (3) and (5).

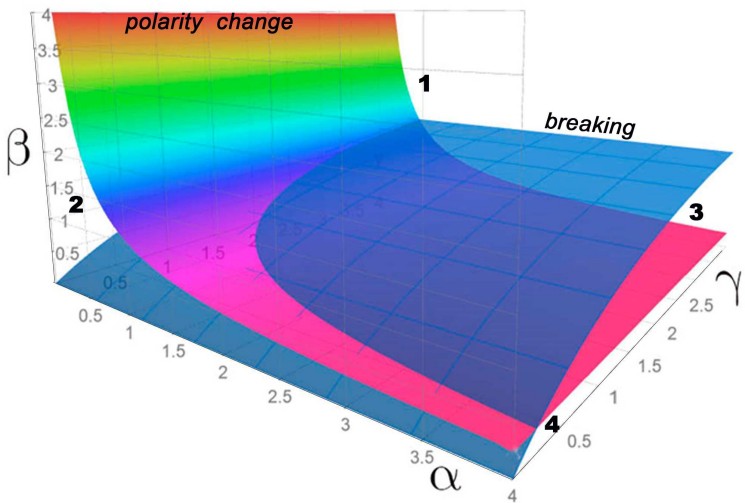

**Figure 2.** Three-dimensional $\alpha, \beta, \gamma$ diagram of regimes (zone 1) without changing polarity and wave breaking, (zone 2) - changing polarity without breaking, (zone 3) regime of wave breaking without changing polarity, (zone 4) - breaking with changing polarity.

115      To compare the $\alpha\beta\gamma$ diagram versus the data from field observations, the results of laboratory measurements and numerical simulations were analysed by Terletska et al. (2020). They are presented in Table 1.

     Terletska et al. (2020) showed that results of field observations (Moum et al., 2003; Vlasenko et al., 2014; New and Pingree, 1990; Navrotsky et al., 2004; Osborn et al., 1980; Orr and Mignerey, 2003; Nam and Send, 2010; Fu et al., 2016), laboratory experiments (Helfrich and Melville, 1986; Cheng et al., 2011) and numerical experiments that simulate ISW transformation in

120 laboratory scales (Talipova et al., 2013; Terletska et al., 2020) are in good agreement with the proposed classification. All data were identified as belonging to the corresponding diagram domain.

**Table 2.** Parameters of ISW in numerical experiments for idealised Lufeng Region in SCS

| $|a_{in}|$(m) | zone | $\alpha$ | $\beta$ | $\gamma$ |
|---|---|---|---|---|
| 20 | 2,4 | 0.4 | 1 | $1°,3°,5°$ |
| 50 | 4 | 1 | 0.4 | $1°,3°,5°$ |
| 100 | 3,4 | 2 | 0.2 | $1°,3°,5°$ |

## 3 Data and methods

Let us consider the transformation of the ISWs in the case of idealised topography and stratification that approximately follows cross-section in the Lufeng Region in SCS. The position of the cross-section is shown in Fig. 1 a. Data indicates (Ramp et al., 2010; Wang et al., 2013; Jackson, 2004) that internal waves from the Luzon strait propagate westward to Dongsha Islands and further to Lufeng Region, and the measured current in waves is about $1.5 - 2.0$ m/s. Wave amplitudes obtained using SAR images (during May after strong thermocline developing in April) at the depth of about 300 m vary from $10$ to $50$ m with the depth of thermocline about $40 - 65$ m (Meng and Zhang, 2003). For numerical modeling of the idealised case that mimics Lufeng Region computational domain with the length $L = 18$ km, and maximal depth $H_{max} = 300$ m was considered (Fig. 1 a). We approximated stratification in the Lufeng Region by the two-layer density profile. The densities of the layers are $\rho_1$ and $\rho_2$ (depths $h_1$ and $h_2$ and $H = h_1 + h_2$.) and the pycnocline layer thickness $dh$ :

$$\rho(z) = \frac{\rho_1 + \rho_2}{2} - \frac{\rho_1 - \rho_2}{2} \tanh\left(\frac{z - h_1}{dh}\right). \tag{7}$$

In numerical experiments we vary waves amplitudes $|a_{in}|$: $|a_{in}| = 20$ m, $|a_{in}| = 50$ m $|a_{in}| = 100$ and slopes $\gamma$: $\gamma = 1^o$, $\gamma = 3^o$, $\gamma = 5^o$. Slope inclination $\gamma$ for smooth curvilinear slope is measured as the maximal slope value. Corresponding values of $\alpha, \beta, \gamma$ are given in the table 2. Density $\rho_1 = 1021.5$ and $\rho_2 = 1025.5$ $(kg/m^3)$ and the pycnocline layer thickness $dh = 15$ m. The flux of salinity through the flume boundaries is also set to zero. Density profile from measurements from SCS (May) (Orr and Mignerey (2003)) and initial density profile (7) are shown in Fig. 3. Depths layers are $h_1 = 50$ m and $h_2 = 250$ m for all runs.

The numerical simulations were carried out using the free-surface non-hydrostatic numerical model (Kanarska and Maderich, 2003; Maderich et al., 2012). The Smagorinsky model extended for the stratified fluid (Siegel and Domaradzki, 1994) was used to explicitly describe the small-scale turbulent mixing and dissipation effects in the ocean scale ISW. Totaly 9 (three $\gamma$ and three $\alpha$ ) runs were carried out for all cases. Spatial resolution was $4.3 \times 1.2 \times 1.2$ m for all cases. Bottom following, the sigma co-ordinate vertical system was used in the present modeling. The quasi-two-dimensional model with a resolution of 4 nodes across the wave tank with the resolution $4200 \times 250$ nodes was used for present calculations. No-slip boundary conditions were applied at the bottom and two end walls. The free-slip conditions were applied at the side walls. Mode-splitting technique

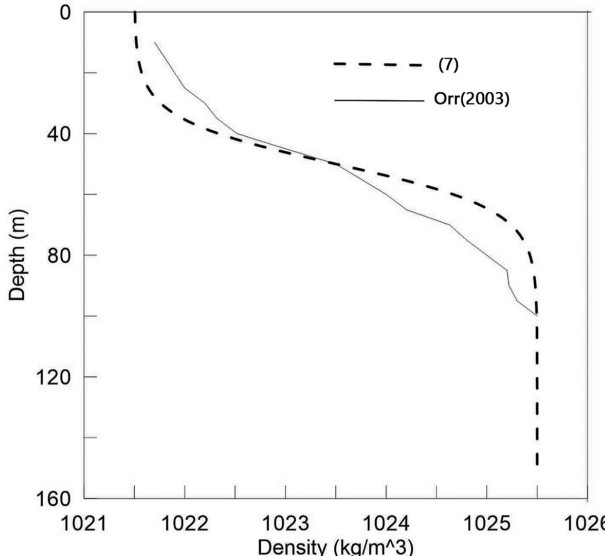

**Figure 3.** Density profile from measurements from SCS (May) (Orr and Mignerey (2003)) and initial density profile (7) used for numerical calculations

and decomposition of pressure and velocity fields on hydrostatic and nonhydrostatic components were used in the numerical method and it is described in detail in (Maderich et al., 2012).

The model was initialized using iterative solution the Dubreil-Jacotin-Long (DJL) (Dubreil-Jacotin, 1932) equation with the initial guess obtained from a weakly nonlinear theory. The DJLES spectral solver from the MATLAB package https://github.com/
150   mdunphy/DJLES/ was used. The transformation of ISW with initial amplitude $|a_{in}| = 50$ m is shown in Fig. 4. The minimum depth of the lower layer over the shelf is $h_{2+} = 20$ m and the slope is $\gamma = 1^o$. The parameters are $\alpha = 1$, $\beta = 0.4$, $\gamma = 1^o$, that corresponds to regime of breaking with changing polarity. ISWs propagation velocity is about 1.2 m/s that is typical for Lufeng Region in SCS (Meng and Zhang, 2003). From (3) and (4) we could find the location on the slope where ISW will change polarity $h_{2+} = h_1 = 50$ m and ISW breaking at the place where $h_{br} \approx 40$ m. It could be seen from Fig. 4 that ISW $|a_{in}| = 50$
155   m at first changes its polarity at time $t = 2$ h 30 min and then breaks at the slope at $t = 2$ h 40 min.

In Fig. 4 a 3D diagram of regimes with the cross-section $\alpha\beta$ for $\gamma = 1^o$ is shown. In the Fig. 5 b red line corresponds to the polarity change criterion (3) whereas the black line corresponds to the breaking criterion (5). Three experiments exp.1: $\alpha = 0.4, \beta = 1, \gamma = 1^o$; exp.2: $\alpha = 1, \beta = 0.4, \gamma = 1^o$; exp.3.: $\alpha = 2, \beta = 0.2, \gamma = 1^o$ are marked. The first one exp.1 represents the cases of interaction of ISW ($|a_{in}| = 50$ m) with polarity change but without breaking. Exp.2 represents the case when the
160   first ISW changes its polarity from depression to elevation types and then breaks 4. Exp.3 is the case when ISW breaks on the slope before it pass the changing polarity point.

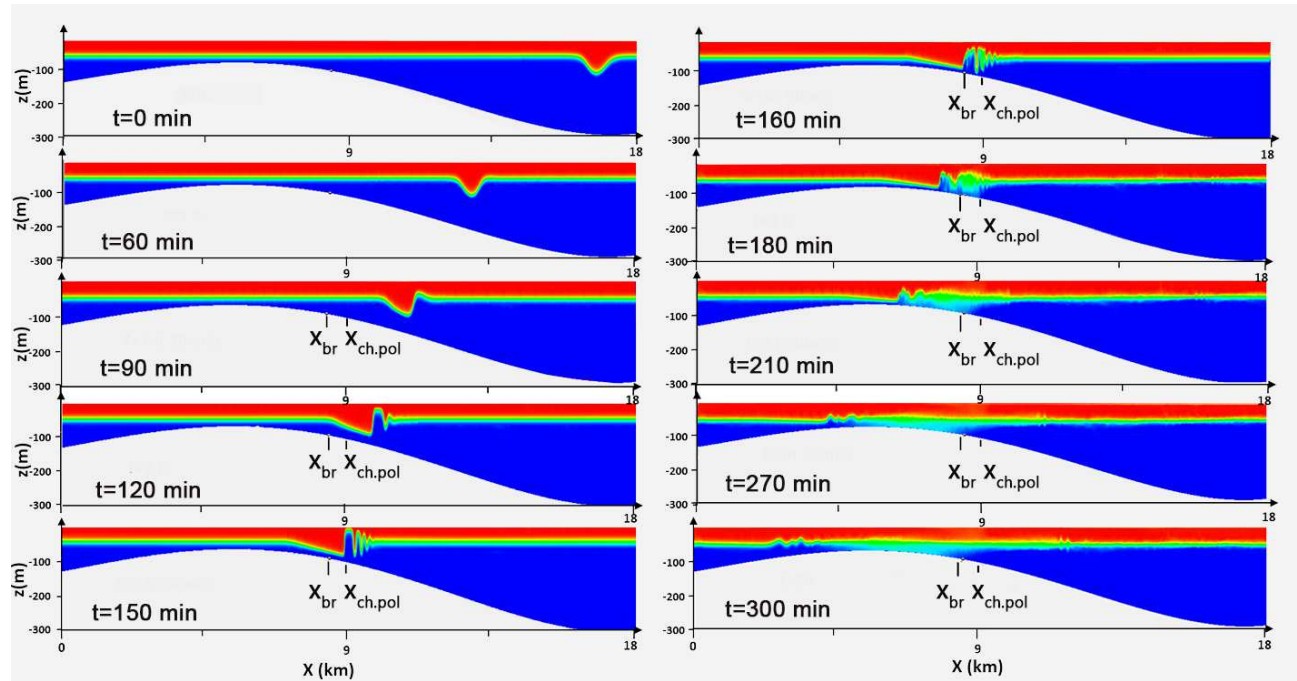

**Figure 4.** The evolution of the ISW with $|a_{in}| = 50$ m in cross-sections at time $t = 0; 60; 90; 120; 150; 160; 180; 210; 270; 300$ min ($\gamma = 1^o$, $\alpha = 1$, $\beta = 0.4$). $x_{br}$ and $x_{ch.pol}$ - are the points of breaking and changing polarity.

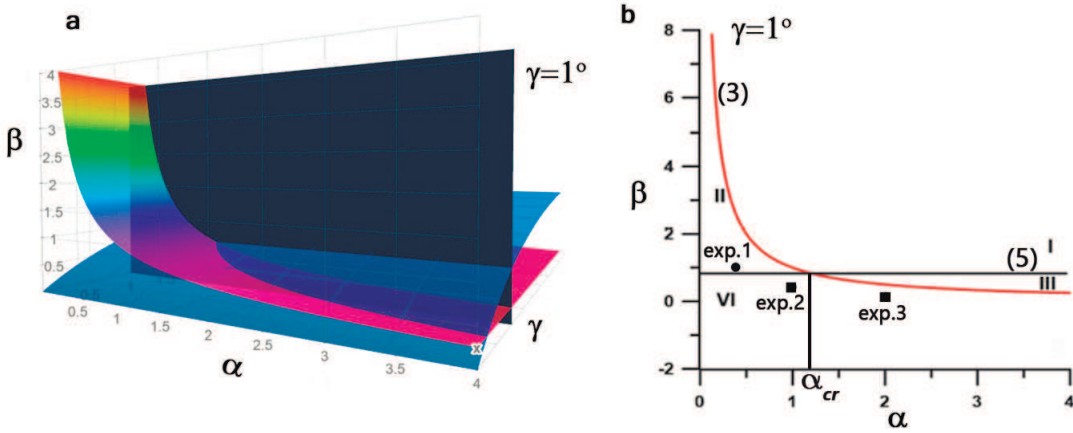

**Figure 5.** 3D diagram of regimes with the cross-section $\alpha\beta$ for $\gamma = 1^o$. The red line corresponds to polarity change criterion (3), black line corresponds to breaking criterion (5). The circles are changing polarity without breaking cases, squares mark cases of changing polarity with breaking.

## 4 Estimate of energy loss in internal waves breaking on slopes

An important characteristic of the wave-slope interaction is the loss of kinetic and available potential energy during the trans-
formation. Energy transformation due to mixing leads to the transition energy to background potential energy and the energy
dissipation. It can be estimated based on the budget of the wave energy before and after the transformation.

It was carried out calculating of energy dissipation for two configurations - for real-scale experiment for idealised Lufeng Re-
gion in SCS and laboratory-scale experiment with the trapezoid shelf-slope configurations (Terletska et al., 2020). Parameters
of ISW in numerical experiments of laboratory-scale experiments by Terletska et al. (2020) are given in Table 3.

The characteristics of the incoming and reflected wave were recorded in the cross-sections $x_r$, which are located near the
foot of the slope, and the wave passing on the shelf, in the cross-section $x_l$ (Fig. 1 b,c).

Energy loss by breaking waves was estimated following Lamb (2007) and Maderich et al. (2010) from the budget of depth-
integrated pseudoenergy. To find the balance of the total energy we have calculated the total energy of the incident, reflected,
and transmitted waves before slope and on the plateau by using the depth-integrated pseudoenergy flux $F(x,t)$

$$F(x,t) = \int_{-H}^{0} (E_{PSE} + p)U dz, \tag{8}$$

where $p$ is pressure disturbance due to passing wave and $U$ is the horizontal velocities. Where $E_{PSE}$ is the pseudoenergy
density, which is a sum of kinetic energy density $E_k$ and available potential density $E_a$ (part of the potential energy available
for conversion into kinetic energy). For calculation of $E_a$ we used reference density profile that was obtained by an adiabatic
rearranging of the density field. Then volume integration of these flows outside the mixing zone allows us to estimate the
energy of the incoming $PSE_{in}$, reflected $PSE_{ref}$, and transmitted on the plateau $PSE_{tr}$ waves, where

$$\begin{aligned}
PSE_{in} &= \int_{x_r}^{L} \int_{-H}^{0} E_{PSE} dz dx = -\int_{t_1}^{t_2} F(x_r,t) dt, \\
PSE_{tr} &= \int_{0}^{x_l} \int_{-H}^{0} E_{PSE} dz dx = -\int_{t_3}^{t_4} F(x_l,t) dt, \\
PSE_{ref} &= \int_{x_l}^{L} \int_{-H}^{0} E_{PSE} dz dx = \int_{t_5}^{t_6} F(x_r,t) dt,
\end{aligned} \tag{9}$$

where $t_1, t_2$ $t_3, t_4$ $t_5, t_6$ are time intervals when incoming, reflected, and transmitted waves passe the given cross-section.

Then the relative estimation of the energy loss $\delta E_{loss}$ is given by

$$\delta E_{loss} = (PSE_{in} - PSE_{tr} - PSE_{ref})/PSE_{in}. \tag{10}$$

where $PSE_{in}$ - is pseudoenergy of the incident wave, $PSE_{tr}$ and $PSE_{ref}$ - is pseudoenergy of transmitted and incident
ones respectively. The energy loss for mixing during the interaction of the wave with the slope of $\delta E_{loss}(\%)$ from the blocking
parameter $\beta$ is shown in fig. 6 a. This field of values $\gamma\beta$ is built by 39 numerical experiments described in table 3, 37 numerical
experiments from Talipova et al. (2013) for $\gamma = 90^o$ and 9 experiments from the present study. $\delta E_{loss}$ was estimated for the

**Table 3.** Parameters of ISW in numerical experiments of laboratory scale [Terletska et al. (2020)].

| $|a_{in}|$(m) | zones | $\alpha$ | $\beta$ | $\gamma$ |
|---|---|---|---|---|
| 0.02 | 2, 4 | 0.25 | 0, 1, 2.5 | $0.5^o, 1.5^\circ, 60^\circ, 90^\circ$ |
| 0.08 | 1, 3, 4 | 1 | 0.3, 1.1, 2.2 | $0.5^o, 1.5^\circ, 60^\circ, 90^\circ$ |
| 0.15 | 1, 3, 4 | 1.5 | 0, 1.5, 2.5 | $0.5^o, 1.5^\circ, 60^\circ, 90^\circ$ |
| 0.15 | 1, 4 | 1.5 | 0.58, 0.8, 1.41 | $1.5^\circ$ |

wide the range of slopes $0.5^o < \gamma < 90^o$ and blocking parameters $-2 < \beta < 8$. ISW energy loss for limiting case of underwater step when $\gamma = 90^o$ was compared with the results of laboratory experiments by Wessels and Hutter (1996) and Chen (2007) ( fig. 6 b). It can be seen that wave transformation in zone 4 is the most dissipative. With this type of transformation, energy losses reach up to $55\%$. For slopes from $5^o < \gamma < 90^o$ dependence of the energy dissipation from the blocking parameter $\beta$ has almost the same pick shape as in the limiting case $\gamma = 90^o$. For mild slopes $\gamma$ we expect an increase of dissipation for all range of blocking parameter values $\beta$.

We can compare the energy dissipation for real scale experiment with the laboratory scale experiment with a close values of $\alpha$ and $\beta$ for slope $\gamma \approx 1$. Consider the cases with $\alpha = 1$ and $\beta = 0.4$ for slope $\gamma = 1^\circ$ (real scale from table 2 ) and $\alpha = 1$ and $\beta = 0.3$ for slope $\gamma = 1.5^\circ$ (laboratory scale experiments table 3 ) (zone 4 - wave breaking regime with polarity change). For strong mixing the difference is about $5\%$ $\delta E_{loss_{r_s}}$ =62 %, $\delta E_{loss_{l_s}}$ =57 %.

To build a zone map for the shelf zone the direction of propagation of internal waves, amplitude of incoming wave and stratification should be defined. They could be found using the approach for estimating the geographic location of high-frequency nonlinear internal waves from (Jackson et al., 2012), amplitudes of the incoming internal waves, the depth of the mixed layer. Fig. 7 shows an example of a map with zones corresponding to the different regimes of interaction described above. These maps were constructed for the case of internal waves with an amplitude of $a_{in} = 50$ m and a mixed layer depth $h_1 = 50$ (Meng and Zhang, 2003). On this map the black line – is isobath 120 m (shelf), the violet line is polarity change curve $h_1 = h_2$ and the red area – is the zone of internal wave breaking where $h_1 + h_{br} > H$.

## 5  Conclusions

A three-dimensional $\alpha\beta\gamma$ classification diagram describing four types of interaction of ISWs with the slopes is discussed. Relations between the parameters $\alpha$, $\beta$, $\gamma$ for each regime were obtained using the empirical relation for wave breaking condition and weakly nonlinear theory for the criterion of changing the polarity of the wave. The distinguished regimes are: (1) ISW propagates over slope without changing polarity and wave breaking; (2) ISW changes polarity over slope without breaking;(3) ISW breaks over slope but without changing polarity; (4) ISW both breaks and change polarity over a slope. The diagram is validated for realistic topography configurations. Numerical modeling of the idealized configuration that mimics the continental shelf at Lufeng Region SCS is carried out. Results numerical experiments that are carried out in the present paper and other

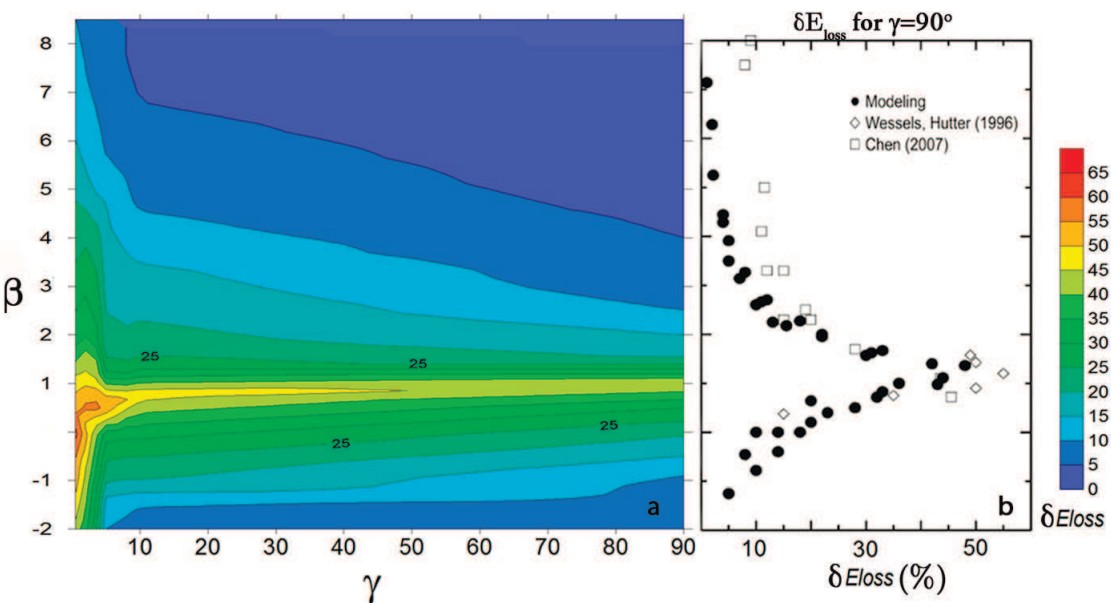

**Figure 6.** Energy loss $\delta E_{loss}$ in internal waves breaking on slopes for $\gamma$ from $0.5°$ to $90°$. For limiting case $\gamma = 90°$ it corresponds to the numerical experiments Talipova et al. (2013) and the results of laboratory experiments Wessels and Hutter (1996) and Chen (2007) with steep obstacles.

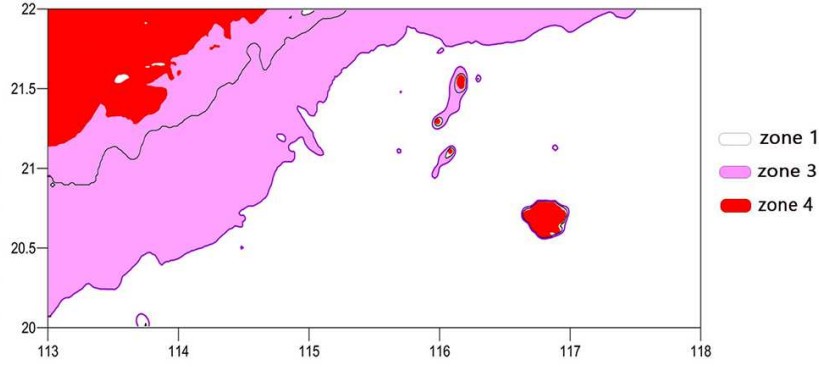

**Figure 7.** Zone map for internal waves transformation over South China Sea shelf with an initial amplitude of 50 m and a depth of the mixed layer 50 m.

laboratory experiments are in good agreement with the proposed classification and estimations. Based on present numerical experiments internal solitary loss of wave energy from transformation over the slope topography is estimated. We concluded

that the results of field, laboratory, and numerical experiments are in good agreement with the proposed classification which can be used for the identification of 'hot spots' of energy dissipation in the ocean.

*Code and data availability.* The output files for all numerical experiments reported in the paper are available from the corresponding author.

*Author contributions.* KT and VM conceived the idea, KT carried out numerical simulations, contributed to the design of figures and participated in the writing of the manuscript. TT commented on the writing of the manuscript contributed large parts of the manuscript

organization and interpretation of the results. VM contributed to writing the manuscript and interpretation the results.

*Competing interests.* The authors declare that they have no conflict of interest.

*Acknowledgements.* This work is partially supported for TT by the Russian Science Foundation (grant No 19-12-00253, Section 2) and the Russian Foundation of Basic Research (grant No 19-05-00161)

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
