# Peer review of "Estimate of energy loss from internal solitary waves breaking on slopes"

_Nonlinear Processes in Geophysics, 2021_

## Referee Comment (RC1)

The authors analyzed evolution of ocean internal solitary wave along a shoaling continental slope using a parameterization analysis method and the continental shelf of the northern South China Sea (SCS) as a test area. The method and results are of a certain value. However, the manuscript was not prepared well for publication.

1. Introduction is redundant and loses logic. It should be re-written after gaining more information.

2. The authors failed in literature hunting, so that they seem not to be familiar with solitary wave theories and progress in research of internal waves in the SCS. Thus, this reviewer recommends the authors at least to read the following publications:

   Wang J., Huang W., Yang J., et al. 2012. Study of the propagation direction of the internal waves in the South China Sea using satellite images. Acta Oceanologica Sinica, 32, 42-50, doi: 10.1007/s13131-013-0312-6.

   Zhao Z. 2014. Internal tide radiation from the Luzon Strait. Journal of Geophysical Research: Oceans, 119, 5434–5448, doi:10.1002/2014JC010014.

   Zheng Q., Susanto R. D., Ho C.-R. et al. 2007. Statistical and dynamical analyses of generation mechanisms of solitary internal waves in the northern South China Sea. Journal of Geophysical Research: Oceans, 112(C03), C03021, doi: https://doi.org/10.1029/2006JC003551.

   Zheng Q. 2017. Satellite SAR Detection of Sub-mesoscale Ocean Dynamic Processes. London: World Scientific, Chapters 6 and 7.

   Zheng (2017) may help knowing comprehensive information on the ocean internal waves. Zheng et al. (2007) may help understanding theories of evolution of internal solitary waves along the shoaling thermocline or topography. Wang et al. (2012) and Zhao (2014) may help understanding that the internal waves may occur anywhere along the continental shelf of the SCS, not only in the Luzon Strait.

   By the way, Junmin (2003) in the manuscript should be Meng and Zhang (2003).

3. Scientific term "wave breaking" was not clearly defined in the manuscript. There would be two different cases. 1) In the case of linear waves, wave breaking means that a wave loses a part of amplitude and becomes a smaller wave. 2) In the case of solitary waves, wave fission means that a large soliton evolves into a packet of smaller solitons. Looking at Figure 4 of the manuscript, one can see that the studied case should belong to the second case. Thus, the energy increment (might not be loss) before and after fission of an incident solitary wave should be calculated by $\Delta E = \sum_{i=1}^{n} E_i - E_0$,
   where $n$ is the total number of small solitary waves after fission, $E_i$ is the total energy of the i-th solitary wave, and $E_0$ is the total energy of the incident large solitary wave.

4. The English writing is far to reach a publishable level. There are too many gramma errors and wrong wordings.

Owing to above severe flaws, this reviewer cannot recommend the manuscript for publication.

---

## Referee Comment (RC2)

In this work, the authors discussed energy loss during the ISW breaking on the slopes. Four types of interaction between ISW and slopes were discussed and a parameterization analysis method was applied in Lufeng region in SCS.

They concluded that the results could be used for the identification of "hot spot" of energy dissipation in the ocean. The results may be valuable for researchers who study the ISW on the slopes. Therefore, I recommend a minor revision for this manuscript.

Detailed comments are listed below:

1. The writing should be improved. There are many grammatical problems in the manuscript making it difficult to read. The text should be revised by a native English-speaking person.

2. The introduction needs to be significantly improved in logic and the key points should be highlighted.

3. A table should be provided in Section 2 for the parameters. The breaking of ISW also requires a clearer definition. In Page 5 Line 16, a number "3" was given but it was meaningless, similar mistakes need to be fixed.

4. More detailed explanations for the configuration of experiments should be given in Section 3. There is a mismatch between Fig. 4 and its caption, the subgraph in Fig. 4 should be marked to remind the readers which one corresponds to a particular moment.

5. How was the pseudo-energy calculated? Please provide the equations.

---

## Author Comment (AC1)

**Response to Reviewer 1**

The authors are most grateful for your comments. We have followed your suggestions and revised the manuscript accordingly. Please, find our responses below marked blue.

The authors analyzed evolution of ocean internal solitary wave along a shoaling continental slope using a parameterization analysis method and the continental shelf of the northern South China Sea (SCS) as a test area. The method and results are of a certain value. However, the manuscript was not prepared well for publication.

1. Introduction is redundant and loses logic. It should be re-written after gaining more information. Answer:

**Answer**: The abstract and introduction were re-written and extended with more information and references. In particular, they were extended to describe the issues of assessing energy losses during the breaking of solitary waves on the shelf.

P. 1 L. 3-6
"The first is the breaking of internal waves over bottom topography when fluid velocities exceed wave phase speed that causes overturning of the rear face of the wave. The second is changing of polarity at the turning point where the depths of upper and lower layers are equal."

P. 2 L. 33-35

"ISW breaking on the slope occurs when fluid velocities in the wave exceed wave phase speed. Leading to the overturning of the rear face of the wave, shear instability, and intensive mixing. Different types of breaking are commonly distinguished by slope inclination, water column stratification, and wave characteristics.

Note that the mechanisms of the generation of these waves are not considered in the article.

2. The authors failed in literature hunting, so that they seem not to be familiar with solitary wave theories and progress in research of internal waves in the SCS. Thus, this reviewer recommends the authors at least to read the following publications:
Wang J., Huang W., Yang J., et al. 2012. Study of the propagation direction of the internal waves in the South China Sea using satellite images. Acta Oceanologica Sinica, 32, 42-50, doi: 10.1007/s13131-013-0312-6.
Zhao Z. 2014. Internal tide radiation from the Luzon Strait. Journal of Geophysical Research: Oceans, 119, 5434–5448, doi:10.1002/2014JC010014.
Zheng Q., Susanto R. D., Ho C.-R. et al. 2007. Statistical and dynamical analyses of generation mechanisms of solitary internal waves in the northern South China Sea. Journal of Geophysical Research: Oceans, 112(C03), C03021, doi: https://doi.org/10.1029/2006JC003551.
Zheng Q. 2017. Satellite SAR Detection of Sub-mesoscale Ocean Dynamic Processes. London: World Scientific, Chapters 6 and 7.
Zheng (2017) may help knowing comprehensive information on the ocean internal waves. Zheng et al. (2007) may help understanding theories of evolution of internal solitary waves along the shoaling thermocline or topography. Wang et al. (2012) and Zhao (2014) may help understanding that the internal waves may occur anywhere along the continental shelf of the SCS, not only in the Luzon Strait.
By the way, Junmin (2003) in the manuscript should be Meng and Zhang (2003).

**Answer:** Thank you for your suggestions. We added information on the internal solitary wave in the South China Sea in the introduction

P. 2 L. 48-54

"Based on the analysis of satellite images [Wang2013, Jackson2004] it was found that in the northeastern South China Sea most internal waves are generated at the Luzon Strait. Further, solitary waves propagate westward and then they diffracted around the Dongsha Islands. In the shallow water regions of the northern SCS, changes in water depth may cause polarity conversion, leading to the transformation of depression ISW to elevation ISWs [Liu1998, Orr2003] showed that the kinetic energy of ISWs decreased three times, after changing polarity, [Zhang 2018] showed that the seasonal variations in stratification cause these seasonal variations in polarity.."

We added information on the observation of ISW in Table 1 on P. 5.
We added 18 new references. The reference Junmin (2003) was corrected as Meng and Zhang (2003).

3. Scientific term "wave breaking" was not clearly defined in the manuscript. There would be two different cases. 1) In the case of linear waves, wave breaking means that a wave loses a part of amplitude and becomes a smaller wave. 2) In the case of solitary waves, wave fission means that a large soliton evolves into a packet of smaller solitons.

**Answer:** we used the term "wave breaking" as the process when fluid velocities in the ISW exceed wave phase speed that causes overturning of the rear face of the wave. Density intrusion or supercritical jets could occur during ISWs breaking for different conditions. We added information to the text.

P. 1 L. 3-6

"The first is the breaking of internal waves over bottom topography when fluid velocities exceed wave phase speed that causes overturning of the rear face of the wave."

P. 2 L. 33-35
"(ii) ISW breaking on the slope occurs when fluid velocities in the wave exceed wave phase speed. That led to the overturning of the rear face of the wave, shear instability, and intensive mixing."

Looking at Figure 4 of the manuscript, one can see that the studied case should belong to the second case. Thus, the energy increment (might not be loss) before and after fission of an incident solitary wave should be calculated by $\Delta E = \sum Eini=1 -E0$, where n is the total number of small solitary waves after fission, Ei is the total energy of the i-th solitary wave, and E0 is the total energy of the incident large solitary wave. 4.

**Answer:**

We estimated energy loss by breaking waves from the budget of depth-integrated pseudoenergy following (Lamb.2007, Terletska 2013). The corresponding text was added:

P. 10. Line 171

"Energy loss by breaking waves was estimated following [Lamb2007] and [Maderich2010] from budget of depth integrated pseudoenergy. To find the balance of the total energy we have calculated the total energy of the incident, reflected, and transmitted waves before slope and on the plateau by using the depth-integrated pseudoenergy flux F(x,t),

$$F(x,t) = \int_{-H}^{0} (E_{PSE} + p) \, U dz$$

where $p$ is pressure disturbance due to passing wave and $U$ is the horizontal velocities. Where $E_{PSE}$ is the pseudoenergy density that is sum of kinetic energy density $Ek$ and available potential density $Ea$ (part of the potential energy available for conversion into kinetic energy). For calculation of $Ea$ we used reference density profile that was obtained by an adiabatic rearranging of the density field. Then volume integration of these flows outside the mixing zone allows us to estimate the energy of the incoming PSEin, reflected PSEref, and transmitted on the plateau PSEtr waves, where

$$\text{PSEin} = \int_{x_r}^{L} \int_{-H}^{0} E_{PSE}\, dz dx \quad , \text{PSEtr} = \int_{0}^{x_l} \int_{-H}^{0} E_{PSE}\, dz dx \quad \text{PSEref} = \int_{x_l}^{L} \int_{-H}^{0} E_{PSE}\, dz dx \quad ."$$

4. The English writing is far to reach a publishable level. There are too many gramma errors and wrong wordings. Owing to above severe flaws, this reviewer cannot recommend the manuscript for publication.

Answer: We have corrected the writing.

---

## Author Comment (AC3)

**Response to Reviewer 2**

The authors are most grateful for your comments. We have followed your suggestions and revised the manuscript accordingly. Please, find our responses below.

In this work, the authors discussed energy loss during the ISW breaking on the slopes. Four types of interaction between ISW and slopes were discussed and a parameterization analysis method was applied in Lufeng region in SCS. They concluded that the results could be used for the identification of "hot spot" of energy dissipation in the ocean. The results may be valuable for researchers who study the ISW on the slopes. Therefore, I recommend a minor revision for this manuscript.

Detailed comments are listed below:

1. The writing should be improved. There are many grammatical problems in the manuscript making it difficult to read. The text should be revised by a native English-speaking person.

Answer: We have revised the writing.

2. The introduction needs to be significantly improved in logic and the key points should be highlighted.

Answer: We rewrite the introduction and added information about internal solitary wave in South China Sea, their characteristics, and add 18 new references.

3. A table should be provided in Section 2 for the parameters.

Answer: We added Table 1 in Section 2 with $\alpha\,\beta\,\gamma$ parameters of ISW in numerical, laboratory experiments and field observations. P. 5. NPG_Terletska_revised.pdf

The breaking of ISW also requires a clearer definition.

Answer: We add the definition of ISW breaking:

P. 1 L. 3-6          NPG_Terletska_revised.pdf
"…. the breaking of internal waves over bottom topography when fluid velocities exceed wave phase speed that causes overturning of the rear face of the wave."

P. 2 L. 33-35          NPG_Terletska_revised.pdf
"(ii) ISW breaking on the slope occurs when fluid velocities in the wave exceed wave phase speed. That led to the overturning of the rear face of the wave, shear instability, and intensive mixing."

In Page 5 Line 16, a number "3" was given but it was meaningless, similar mistakes need to be fixed.

Answer: Thank you. We fixed mistakes and other typos

4. More detailed explanations for the configuration of experiments should be given in Section 3.

Answer: The text was added

P.7 line 136.   NPG_Terletska_revised.pdf

"The flux of salinity through the flume boundaries is also set to zero."

Also we modify Tables 2 and 3 by adding the corresponding column with zones for each set of the experiment.

There is a mismatch between Fig. 4 and its caption, the subgraph in Fig. 4 should be marked to remind the readers which one corresponds to a particular moment.

Answer:  The caption to the Fig. 4 was corrected. The time for each snapshot in Fig.4 was changed accordingly with the caption.

5. How was the pseudo-energy calculated? Please provide the equations.

Answer: We add the equation for the pseudo-energy calculation in the text

P. 10. Line 171   NPG_Terletska_revised.pdf

"Energy loss by breaking waves was estimated following [Lamb2007] and [Maderich2010] from budget of depth integrated pseudoenergy. To find the balance of the total energy we have calculated the total energy of the incident, reflected, and transmitted waves before slope and on the plateau by using the depth-integrated pseudoenergy flux F(x,t),

$$F(x,t)=\int_{-H}^{0}(E_{PSE} + p)\, U dz$$

where $p$ is pressure disturbance due to passing wave and $U$ is the horizontal velocities. Where $E_{PSE}$ is the pseudoenergy density that is sum of kinetic energy density $Ek$ and available potential density $Ea$ (part of the potential energy available for conversion into kinetic energy). For calculation of $Ea$ we used reference density profile that was obtained by an adiabatic rearranging of the density field.  Then volume integration of these flows outside the mixing zone allows us to estimate the energy of the incoming PSEin, reflected PSEref, and transmitted on the plateau PSEtr waves, where

$$PSEin =\int_{x_r}^{L} \int_{-H}^{0} E_{PSE}\, dzdx \quad, PSEtr =\int_{0}^{x_l} \int_{-H}^{0} E_{PSE}\, dzdx \quad PSEref =\int_{x_l}^{L} \int_{-H}^{0} E_{PSE}\, dzdx \quad."$$